# Comparison of sampling methods for small oxbow wetland fish communities

**Dylan M. Osterhaus** [1¤], **Samuel S. Leberg**[1], **Clay L. Pierce**[1], **Timothy W. Stewart** [1]*,
**Audrey McCombs**[2]

**1** Department of Natural Resources Ecology and Management, Iowa State University, Ames, Iowa, United States of America, **2** Department of Statistics, Iowa State University, Ames, Iowa, United States of America

¤ Current address: Department of Biology, New Mexico State University, Las Cruces, New Mexico, United States of America

* twstewar@iastate.edu

**Data Availability Statement:** All data are available in figshare repository (DOI 10.6084/m9.figshare. 21395553).

**Funding:** Funding for this project was provided by the Iowa Soybean Association and Syngenta Crop

## Abstract

Throughout the world, wetlands have experienced degradation and declines in areal coverage. Fortunately, recognition of the value of wetlands has generated interest in preserving and restoring them. Post-restoration monitoring is necessary to analyze success or failure, thereby informing subsequent management decisions. Restoration of oxbow wetlands has become the focus of targeted restoration efforts to promote recovery of biodiversity and sensitive species, and to enhance ecosystem services. The fish communities of oxbows have been the subject of many monitoring studies. However, a recommended sampling methodology for monitoring the fish communities of oxbows has not been described, thereby limiting our capacity to effectively monitor these ecosystems. We compared four sampling methodologies (backpack electrofishing, fyke netting, minnow trapping, and seining) for fish community data collection with a primary objective of determining an effective method for sampling fish communities in small oxbow wetlands. Seining and fyke netting were determined to be effective methods for sampling oxbow fish communities. Backpack electrofishing and minnow trapping produced low taxonomic richness values and sampled a smaller proportion of species present than seining and fyke netting. Although seining and fyke netting produced similar taxonomic diversity and abundance values, these two gears differ in their ease of implementation and potential habitat disturbance generated by sampling. Therefore, consideration must be given to how species present (especially sensitive species) within the wetland could be impacted by sampling disturbance when choosing between seining and fyke netting.

## Introduction

Wetlands are among the most impacted aquatic ecosystems, with declines in both areal coverage and habitat quality occurring as the global human population increases [1]. Fortunately, recognition of the value of wetlands has generated interest in preserving and restoring them [2]. Wetland restoration activities range in scope and complexity from watershed-scale undertakings to targeted fine-scale enhancements of habitats. As these activities have expanded, so

Protection. The funders of this study had no role in study design, data collection and analysis, decision to publish, or preparation of the manuscript.

**Competing interests:** The authors have declared that no competing interests exist.

has realization of the value of monitoring oxbows post-restoration to assess restoration successes or failures [3]. Identifying preferred methods of biological monitoring are imperative for comparing success of different restoration strategies and responses of different types of wetlands to these management actions [3].

Within the Midwestern United States, oxbow wetlands (hereafter oxbows) have recently become the focus of targeted restoration efforts due to the habitat that they provide for endangered Topeka Shiner *Notropis topeka*, and various other ecosystem services [4–7]. Oxbows are floodplain habitats consisting of river meanders that have been disconnected from the main channel through natural erosional processes, creating an off-channel lentic habitat [8, 9]. These oxbow habitats are periodically reconnected with the main stream channel during high flow events which allows for colonization of oxbows by fish and dispersal of fishes from the oxbow back into the stream. As prairies, forests, and other forms of natural land cover in the region were replaced by cropland and pasture, streams were channelized, eliminating many oxbows [10, 11]. As oxbows declined in number, their ecosystem services were also lost. In floodplain habitats, oxbows are especially valuable because they retain water, thereby reducing downstream flooding [5]. By sequestering nutrients, oxbows can contribute to improved stream water quality [5, 6]. Additionally, oxbows contribute to increased biodiversity by providing essential wetland habitat for many species [6, 12–14].

Restoration of oxbows has increased substantially over the last twenty years within the states of Iowa, Minnesota, and South Dakota. In Iowa alone, restoration of oxbows has increased from 1.2 restorations per year from 2001–2005 to 12.8 restorations per year from 2016–2020 [15]. Iowa oxbows are being restored because of ecosystem services described above, and because they appear to provide critically important habitat for the Topeka Shiner [4–7].

Accelerating oxbow restoration has led to increased monitoring of these ecosystems by The Nature Conservancy, the United States Fish and Wildlife Service, the United States Geological Survey and other entities. Monitoring projects are conducted in part to determine success or failure of restoring oxbow ecosystem services and habitat for Topeka Shiner [16]. Monitoring typically involves collecting habitat data (depth, width, length, bank angles, substrate, canopy cover, etc.) and conducting plant and invertebrate community surveys. Given that oxbows are critical habitat for Topeka Shiner, research has also focused on fish communities of restored and unrestored oxbows [7, 17–19].

While many studies have examined oxbow fish communities, best sampling practices for these habitats are not known, and some methods are likely more effective than others [17]. Additionally, given the differences in habitat morphology between oxbows and typical stream channels or wetlands, sampling methods that are more appropriate for streams or wetlands may not function as well when employed within oxbows. Sampling methodologies employed in previous oxbow fish community surveys have been varied and included seining with a bag seine of variable sizes and numbers of seine passes [17], blocking the oxbow into four sections and sampling three while leaving one section as an undisturbed fish refuge [18], and "scoop" seining (1.2 x 1.2 m seine with 6.4 mm mesh) [20]. Because the Topeka Shiner is a rarely encountered species with a restricted geographic distribution and specific habitat requirements, it is important to identify best practices for fish community survey methods for oxbows that are restored for Topeka Shiner habitat. An effective survey methodology would produce consistently high catch rates, accurate information for species diversity and would minimize potential stress to fish and habitat. Determination and implementation of best sampling methodologies for future studies will enable sampling without causing significant fish mortality, and efficient/effective sampling of the fish community.

In this study, we quantitatively compared values for four fish community metrics, CPUA (catch per unit area; number of fishes per 100 $m^2$ oxbow surface area), species richness, 10th-

90[th] percentile length ranges of fish collected, and proportion of available species pool sampled that were obtained using four sampling methods. Additionally, we provide commentary on sampling considerations pertaining to ease of use, potential handling stress and mortality to fish, habitat disturbance, ability to sample various oxbow habitats and ability to sample in deep water for each of the top determined sampling methodologies analyzed. The objective of our study is to compare performance of sampling gears for oxbow fish communities and given the focus of oxbow restoration on Topeka Shiner recovery, to make recommendations for appropriate sampling gears with consideration for sampling which may involve sensitive or endangered species.

## Materials and methods

Study oxbows consisted of 12 recently restored oxbows located in central Iowa (10 within the Boone River basin, two within the North Raccoon River basin). Oxbow length was measured along the centerline of the oxbow at points equidistant from each bank and ranged from 23 m to 139 m with an average of 77.4 m across oxbows. Oxbow width was measured as the wetted width at eight evenly spaced points along the length of the oxbow. These measurements were averaged at each oxbow and average oxbow width ranged from 4.8 m to 20.4 m with an average of 10.0 m across oxbows. Depth was measured at three points along eight evenly spaced transects spanning the wetted width of the oxbow. These measurements were averaged at each oxbow and average oxbow depth ranged from 0.56 m to 0.98 m with an average of 0.69 m across oxbows. Oxbow surface area was calculated by multiplying the average width of the oxbow by the length.

Fish communities at all oxbows were surveyed over a four-week period (18 May 2021–19 June 2021) using two active (backpack electrofishing, seining) and two passive (minnow trapping, fyke netting) sampling methodologies. Each sampling methodology was used once at each oxbow for a total of 4 sampling events at each oxbow. Seining is a frequently used fish sampling methodology in oxbows [7, 17]. Fyke netting and minnow trapping are commonly used to sample fish in a variety of depressional wetlands [21–24]. Although infrequently used in wetlands, we included backpack electrofishing given the widespread use of this sampling method in stream surveys.

Oxbow sampling sequence and order of methodology implementation were randomized (using a random number generator) to control for time effects on results. For each sampling method, all fish collected were identified to species and total numbers were counted. Total lengths of the first 50 recovered individuals for each species were measured. All fishes were subsequently released into the oxbow from which they were captured. All sampling methods were approved by the Iowa State University Institutional Animal Care and Use Committee (permit number IACUC-20-190).

### Sampling methods

**Active gears.** Backpack electrofishing was performed by a single pass with a backpack electrofishing unit (Smith-Root LR-20B Backpack Electrofisher) across all wadeable habitat (< 1m in depth) by walking along the entire shoreline perimeter. One investigator carried the backpack, while two netters (one on each side of the backpack) collected fish. Electrofishing settings were determined based on conductivity of water in the oxbow (10–500 μS/cm, 200–300 volts; 500–800 μS/cm, 150–200 volts; 800–1000 μS/cm, 120–180 volts; >1000 μS/cm, 100–150 volts).

Seining was conducted using a single pass of a bag seine (10.7 m×1.8 m, 0.6 cm mesh or 16.8 m×1.8 m, 0.6 cm mesh depending on oxbow width) along the entire length of the oxbow.

For oxbows < 16 m in average width, the 10.7 m wide seine was used. For oxbows >16 m in average width, the 16.8 m wide seine was used. A single seine pass (rather than multiple passes) has been determined to be sufficient for surveying the fish community of an oxbow [7]. A single seine pass is more desirable compared to multiple seine passes given the reduced effort, and decreased fish stress and mortality [17].

**Passive gears.** Due to gear limitations, constant effort was employed for both minnow trapping and fyke netting for this study. Future studies should analyze how variable effort, scaled to oxbow size, impacts the sampling performance of both of these passive methods. Minnow trapping consisted of placing four, un-baited, minnow traps (2.5 cm opening, 0.6 cm mesh, 0.4 m long × 0.2 m diameter) at points evenly spaced along the length of the oxbow and at variable depths (0.3 m to 1.1 m) with two traps on each bank. Minnow traps were retrieved after 24 h of deployment.

Fyke netting consisted of placing three, un-baited, mini-fyke nets (4.0 m lead, 0.6 cm mesh, largest hoop opening = 0.6 m×1.2 m) in each oxbow. Previous studies in depressional wetlands determined that 3 fyke nets were sufficient for determining relationships between fish community data, abundance and taxon richness of plants and invertebrates, and physical attributes of the wetland [24, 25]. Nets were placed perpendicular to the shoreline within the open water zone at points spaced evenly along the length of the oxbow. Fyke nets were deployed for 24 h, and then retrieved.

## Fish community metrics and data analysis

To investigate effects of sampling method, we focused our quantitative analyses on four fish community metrics (CPUA, species richness, $10^{th}$–$90^{th}$ percentile range lengths, proportion of available species sampled). Given the variability in sizes of oxbows sampled catch must be scaled to area to be comparable between oxbows. Additionally, given the "closed" nature of these oxbow systems we chose to calculate CPUA in terms of total oxbow surface area (number of fish collected per 100 $m^2$ of oxbow surface area), rather than amount of area sampled or sampling effort. Given that the goal of sampling for our study was to document the entire fish community present within these closed systems at the time of sampling, we believe this to be the best metric to compare sampling methods in terms of fish catch as this metric considers all gears as having sampled the entirety of the oxbow. Species richness was quantified as the number of species collected from one oxbow during one sampling event. The 10-$90^{th}$ percentile range for lengths was calculated for each sampling event and these values were used for comparison. We considered the available species pool at each oxbow as the total species richness as sampled across all four methods at the oxbow. We then calculated the proportion of available species sampled at each oxbow for each gear type by dividing the number of species sampled by a gear by the total species sampled at the oxbow across all gears. QQ-plots and histograms indicated that values for species richness and CPUA were non-normal and heteroscedastic, therefore, these values were log transformed prior to analysis to meet normality and heteroscedasticity assumptions for associated analyses.

All analyses and figures were conducted and created using program R (version 3.6.3) [26] and packages **emmeans**, **ggridges**, **vegan**, **ggplot2**, **lme4**, and **plyr**. Mixed effects linear regression models were fit for each quantitative metric of interest (species richness, CPUA, $10^{th}$-$90^{th}$ percentile length range) with oxbow as a random effect and sampling method, and week of the sampling period included as additive predictors. A type II ANOVA was used to test for effects of sampling method on each fish community metric. The emmeans function was used to examine contrasts between sampling methods for each metric with results averaged over the oxbows and weeks and P values adjusted using the Tukey method for comparing a family of

four estimates. Result of a comparison was considered statistically significant if the P-value of the test was less than or equal to 0.05.

## Results

In total, 48 sampling events occurred (12 oxbows x 4 sampling events at each oxbow), resulting in collection of 32862 individual fish and 26 species (For a summary of fish catch data, refer to S1 Table). Within the sampling period, a fish-kill occurred at one oxbow, likely due to low dissolved oxygen levels as a result of decaying organic matter. Therefore, data from that oxbow were not included in analysis, resulting in 11 oxbows and 44 sampling events included in statistical analysis.

Fathead Minnow *Pimephales promelas* was collected from all oxbows and represented 73% of the fish collected during sampling. Golden Shiner *Notemigonus crysoleucas* was the second most common fish sampled, representing nearly 9% of the total catch, although this species was collected from only 5 of the 11 oxbows. Seven of the 26 species collected represented over 97% of the total fish collected (S1 Table). Five species were collected from only one oxbow while two species were collected at each oxbow (S1 Table). Two Topeka Shiner were collected during sampling.

Sampling method had a significant effect on CPUA ($\chi^2 = 95.11$, df = 3, P < 0.001), species richness ($\chi^2 = 52.29$, df = 3, P < 0.001), fish lengths ($\chi^2 = 21.77$, df = 3, P < 0.002; Fig 1), and proportion of available species sampled ($\chi^2 = 63.14$, df = 3, P < 0.001). CPUA was significantly greater for fyke netting and seining (293.65 ± 469.51 and 449.37 ± 630.99 respectively; average ± SD) than backpack electrofishing (22.79 ± 35.79) and minnow trapping (47.14 ± 99.90; Fig 1; P < 0.001). Species richness was significantly greater for fyke netting (6.55 ± 4.23; average ± SD), seining (7.36 ± 4.82) and backpack electrofishing (4.18 ± 3.03) than minnow trapping (1.45 ± 0.82; Fig 1; P ≤ 0.009). The range of fish lengths collected by seining (81.88 ± 40.75; average ± SD), fyke netting (75.65 ± 37.51), and backpack electrofishing (59.93 ± 34.81) were significantly greater than the range of fish lengths collected by minnow trapping (60.86 ± 13.87; Fig 1; P < 0.04). Furthermore, across all oxbows, seining and fyke netting each collected 24 of the 26 total species encountered while electrofishing and minnow trapping collected only 19 and 7 species respectively. Seining and fyke netting collected a higher proportion of the total species pool across oxbows (0.79 ± 0.21 and 0.72 ± 0.18 respectively; average ± SD), than did backpack electrofishing (0.45 ± 0.20) and minnow trapping (0.21 ± 0.13; Fig 2; P < 0.001).

### Backpack electrofishing

CPUA values obtained via backpack electrofishing were low compared to other methods, indicating that this method does not perform well in sampling oxbow fish communities (Fig 1). Species richness values produced by backpack electrofishing (range = 1–11 species per oxbow based on 11 sampling events) were lower than fyke netting (2–16 species) and seining (2–16 species), indicating that backpack electrofishing fails to detect many species occupying an oxbow (Table 1). Backpack electrofishing was successful in collecting the 13 most common of the 26 total species collected (S1 Table). However, seven of the 13 least common species were absent from collections made via backpack electrofishing. Additionally, on average, backpack electrofishing collected only half of the species pool present at each oxbows (Fig 2). Length frequency distribution data indicate that backpack electrofishing samples fish similarly to fyke netting and seining and collects a larger range of fishes than does minnow trapping (Figs 1 and 3).

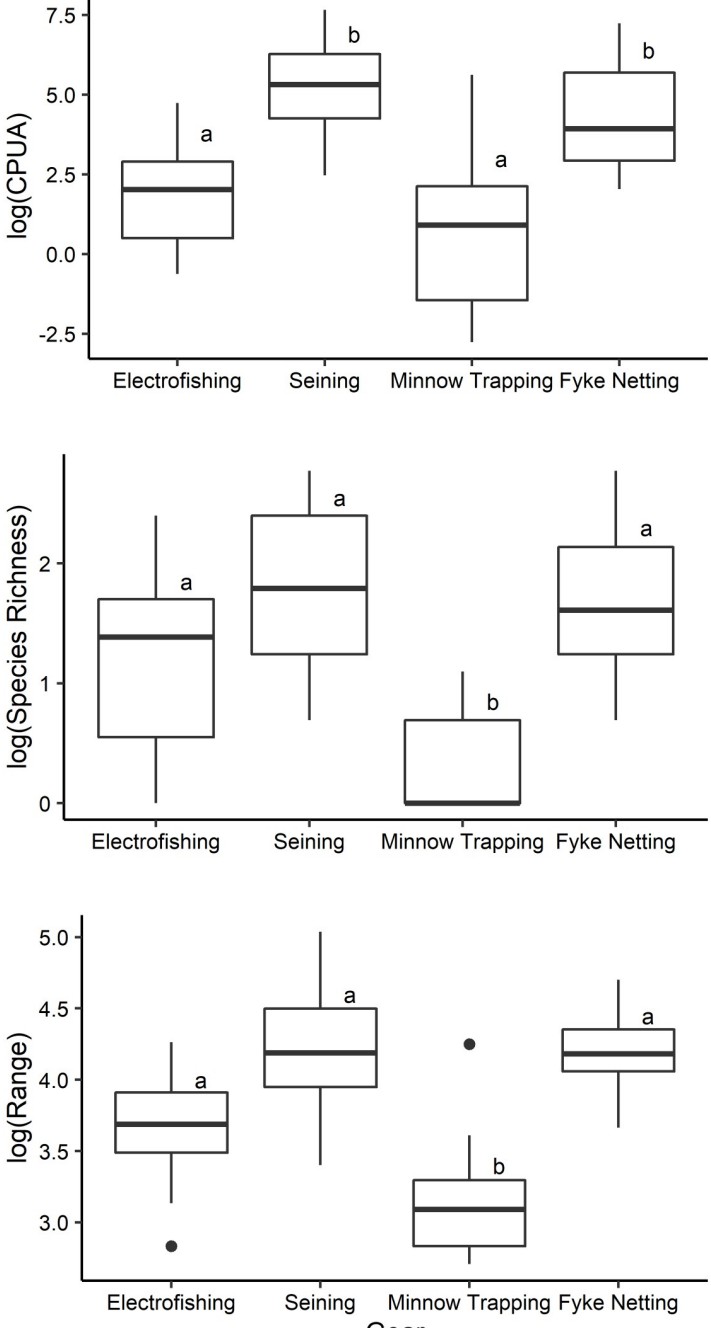

**Fig 1. Boxplots for CPUA (top panel), species richness (middle panel), and 10th-90th percentile length ranges (bottom panel) across all oxbows as a function of sampling method.** Letters denote significant differences between sampling methods.

## Fyke netting

CPUA values from fyke netting were comparatively high, indicating that this technique performs well in capturing fishes occupying the oxbow (Fig 1). Furthermore, fyke netting resulted in the collection of 24 of the 26 fish species recorded during the study, missing only Rock Bass

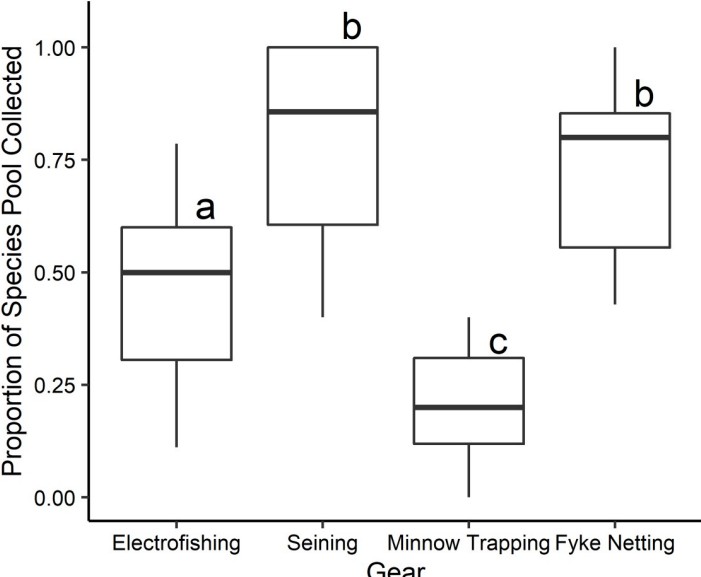

**Fig 2. Boxplots for proportion of species pool collected across oxbows for each sampling method.** Letters denote significant differences between sampling methods.

(representing the lowest proportion of total catch, collected at one oxbow via seining) and Yellow Perch (representing the fourth lowest proportion of total catch, collected at one oxbow via seining) (S1 Table). Across all oxbows, fyke netting sampled a higher proportion of species available at each oxbows (Fig 2). Additionally, a single Topeka Shiner was collected via fyke netting at one oxbow. These findings, in combination with the high species richness values, indicate that fyke netting is effective at detecting rarer species. Length frequency distribution results from fyke netting indicate that this method samples a greater range of fish sizes present in the oxbow compared to other methods used (Figs 1 and 3).

**Table 1. Descriptive statistics by sampling method for each quantitative metric.**

| Sampling Method | Species Richness | CPUA (no. fish per 100 m$^2$) | Fish length (mm) |
|---|---|---|---|
| Backpack Electrofishing | | | |
| mean ± SD | 4.18 ± 3.03 | 22.79 ± 35.79 | 59.93 ± 34.81 |
| Range | 1–11 | 0.62–113.82 | 12–282 |
| Fyke Netting | | | |
| mean ± SD | 6.55 ± 4.23 | 293.65 ± 469.51 | 75.65 ± 37.51 |
| Range | 2–16 | 7.64–1383.28 | 21–375 |
| Minnow Trapping | | | |
| mean ± SD | 1.45 ± 0.82 | 47.15 ± 99.90 | 60.86 ± 13.87 |
| Range | 0–3 | 0–276.92 | 25–118 |
| Seining | | | |
| mean ± SD | 7.36 ± 4.82 | 449.37 ± 630.99 | 81.88 ± 40.75 |
| Range | 2–16 | 11.77–2114.04 | 15–360 |

Overall mean (± SD) and range values are based on 11 samples (11 wetlands x 1 sampling events per wetland).

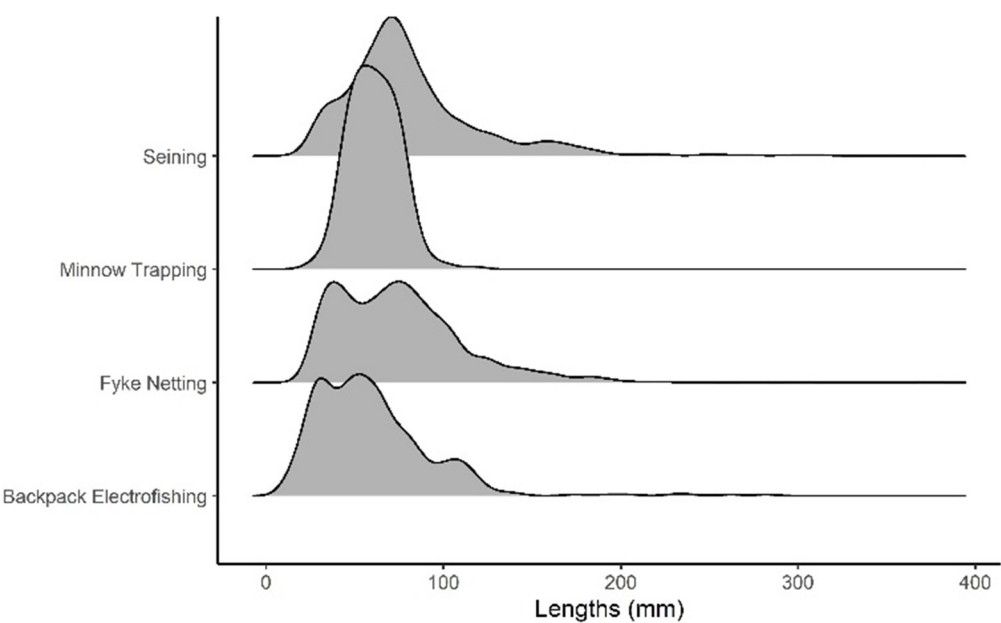

**Fig 3. Length frequency distribution for lengths as sampled by each sampling method (n = 1716, 215, 1344, and 491) respectively for seining, minnow trapping, fyke netting, and backpack electrofishing.** Height of the shaded region for each sampling method along y-axis represents the frequency of fishes of that length.

## Minnow trapping

CPUA values from minnow trapping were the lowest of the four methods tested (Fig 1). Minnow trapping resulted in collection of seven species in total, but no more than three species were collected during any one sampling event, and a small proportion of the available species pool was sampled across oxbows (Table 1, Fig 2). Length ranges of fish collected via minnow trapping were significantly smaller than those collected via backpack electrofishing, fyke netting, and seining (Fig 1). Minnow trapping appears to be more size-selective for fish in the length range of 40 to 90 mm, in comparison to other sampling methods (Figs 1 and 3).

## Seining

CPUA values indicate that sampling performance is high for seining. Seining resulted in collection of 24 of the 26 species found during this study, missing only Black Crappie (tenth lowest proportion of total catch, collected at one oxbow via fyke netting) and Carmine Shiner (third lowest proportion of total catch, collected at one oxbow via fyke netting; S1 Table). Across all oxbows, seining sampled a higher proportion of species available at each oxbows (Fig 2). The range of species richness values generated by seining (2–16 species) was identical to that of fyke netting (Table 1). Furthermore, a single Topeka Shiner was collected via seining at one oxbow. As was true for fyke netting, it appears that seining effectively samples rarer species of fish. Seining does not appear to be size-selective (Figs 1 and 3).

## Discussion

### Comparison of sampling methods for fish community sampling

**Active gears.** Backpack electrofishing is seemingly ineffective in accurately describing the fish community of an oxbow in terms of CPUA and species richness as values for each of these metrics were low relative to other methods across all oxbows. Furthermore, backpack

electrofishing failed to collect many of the rarer species present within the oxbow habitats that were collected by other methods. This result may be unsurprising given the potential drawbacks of using this sampling method in oxbow habitats. First and foremost, turbidity is typically high within oxbows and sediment is easily suspended. Given that efficacy of backpack electrofishing relies heavily on the netter's ability to see stunned fish [27], relying on this sampling method in turbid systems is not recommended. Furthermore, the sampled oxbows frequently contained areas in which water depth was > 1m. To maintain the backpack electrofishing unit's functionality by avoiding the submersion of the circuitry and power source, much deep-water habitat was unsampled. Presence of dense submerged plants also created difficulty because fish stunned by the electrical currents became ensnared in vegetation and did not surface or were simply too entangled in vegetation to be netted. Given the poor performance of backpack electrofishing in comparison to other sampling methods tested, we do not recommend the use of this method for sampling oxbow fish communities.

Across all oxbows, seining resulted in higher CPUA and species richness values in comparison to backpack electrofishing using the methods we employed and was effective in sampling rarer species. Given the morphology of typical oxbows (U-shaped channel, average depths typically < 2 m), in oxbows with limited amounts of aquatic vegetation or obstructions, seining can effectively sample the fish community in all available habitat. However, in oxbows with significant amounts of aquatic vegetation or obstructions, seining can be impeded as the net fills with vegetation or obstructions must be avoided. Given the ability to effectively sample the fish community within all available habitat in oxbows, we recommend seining as a sampling method for oxbow fish communities.

**Passive gears.** Minnow trapping is seemingly ineffective in accurately describing the oxbow fish community using the methods we employed given the consistently low CPUA and species richness values we found across all oxbows. Minnow trapping failed to collect many species that were collected by other methods implemented. As we did not scale the number of minnow traps employed to oxbow size, it is possible that under sampling occurred. However, it is unlikely that increasing the number of minnow traps would greatly improve the performance of this gear type. Multiple other studies have determined that the catch of minnow traps is biased towards common species, is size-selective, and performs poorly in comparison to other methods; our findings corroborate these studies [28–30]. Given the inability to sample the fish community effectively and unbiasedly (as determined by our study and others), we do not recommend use of minnow trapping as a sampling method for oxbow fish communities.

Fyke netting resulted in higher CPUA and species richness values in comparison to minnow trapping using the methods we employed. Additionally, as was true for seining, fyke netting was effective in sampling rarer species within oxbows. However, similarly to minnow trapping, it is possible that under sampling of larger oxbows and over sampling of smaller oxbows occurred as the number of fyke nets was not scaled to oxbow size. We believe this is unlikely as, fyke netting performed similarly to seining (a gear which samples the entirety of the oxbow in an unbiased fashion) in terms of CPUA, species richness, and proportion of available species sampled across all oxbows regardless of size. However, future studies should analyze scaling the number of fyke nets to oxbow size to determine an optimal number of fyke nets given an oxbows surface area. An analysis such as this could further improve fyke netting as a method for sampling oxbow fish communities.

## Gear recommendations and sampling considerations

Overall, seining performed better than backpack electrofishing and fyke netting performed better than minnow trapping. When compared to each other, seining and fyke netting were

comparable in how they sample oxbow fish communities in terms of CPUA, species richness and proportion of species pool collected across oxbows (Table 1, Fig 2), unbiased sampling of various size classes (Fig 3), and ability to sample deep-water habitat. Additionally, with the emphasis of many oxbow monitoring projects placed on Topeka Shiner conservation, it is also important to note that both seining and fyke netting performed well in collecting species that were rare across oxbows, and a high proportion of the available species pool (Fig 2). Therefore, we recommend both seining and fyke netting as best sampling methods for oxbow fish communities. However, while fyke netting and seining both perform well in sampling oxbow fish communities, there are differences in various aspects of sampling (including ease of implementation and how the gear interacts with habitat and fishes) that must be considered when choosing between these gear types.

**Ease of implementation.** When choosing between fyke netting and seining, consideration must be given to the amount of time needed to conduct sampling, habitat characteristics (such as aquatic vegetation and oxbow morphology), and accessibility of the oxbow. For studies constrained by a short timeline, seining may be more desirable as an oxbow can be visited once and sampled via seine within a few hours, while fyke netting may require two days and two visits to the oxbow to complete. While it is possible that fyke netting may be able to effectively sample the fish community of oxbow habitats using a deployment time less than what was implemented in this study (24 hrs), it is unlikely that this time may be decreased enough to be similar to the time investment of seining. Future studies should analyze optimal deployment time (as well as number of nets as previously mentioned) to further optimize the use of fyke netting to sample oxbow fish communities.

Regarding oxbow habitat characteristics, for oxbows containing large quantities of aquatic vegetation, woody debris, or boulders, fyke netting is the preferred sampling method given the difficulty in seining through aquatic vegetation and obstructions [31]. In oxbows with vegetation and obstructions, fyke nets may be placed in ways to avoid these obstructions, while seining may result in habitat remaining unsampled as obstructions and vegetation must be avoided. Furthermore, oxbow morphology (width and depth) must be considered when choosing between seining or fyke netting. Depending on the size (width) of seine available, fyke netting may be preferable in oxbows that are of a width that is significantly greater than that of the seine. Additionally, if the depth of the oxbow is greater than that of the height of the seine net, fishes may escape from the net as they pass over the top of the net or underneath the lead line. Given the variability of morphology and habitat characteristics between oxbows, visiting oxbows prior to sampling may be necessary to inform decisions pertaining to which sampling method to employ.

Fyke netting and seining also differ in amount and weight of gear needed. Fyke netting requires multiple nets, each of which are bulky and may be heavier than a seine depending on net type. However, if seining is employed, only one piece of gear is needed, and this gear can be transported easily with one or two individuals. Therefore, for oxbows that are easily accessible (can be approached closely with a vehicle), both seining and fyke netting can easily be implemented. However, for oxbows that are not easily accessible, seining may be preferable given the ease by which gear can be transported to the oxbows.

**Interaction with habitat and sampled fishes.** In addition to sampling method performance, consideration must be given to potential stress and mortality experienced by fish (either directly via handling or indirectly via habitat disturbance) and sensitivity of potentially encountered species to habitat disturbance and handling (including consideration of spawning season) as surveys are conducted. These considerations are especially significant when handling species of greatest conservation concern [32]. In terms of potential disturbance to habitat, seining requires at least two individuals to walk through the perimeter of the oxbow, and a

net with a lead line to be pulled through the entirety of the oxbow. As a result, habitat disturbance can be significant as aquatic vegetation is displaced by the net and those operating the gear [33]. However, fyke netting only requires the placement of nets at a few specific locations within the oxbow, leading to habitat disturbance to a much smaller extent.

For studies sampling oxbows which may contain species sensitive to habitat disturbance, fyke netting may be preferable to seining given the decreased habitat disturbance of this gear and special considerations should be given to timing of the spawning season for the species. For example, if sampling occurs during the breeding season for nest building centrarchid sunfishes such as the Green Sunfish, and Bluegill (Topeka Shiner nest in association with each of these species) [20, 34–36], this reproductive activity could be compromised if the nests of the centrarchids are destroyed by seining. Therefore, fyke netting is recommended as the preferred sampling method during the Topeka Shiner breeding season.

In addition to the indirect disturbances caused by sampling with seining and fyke netting, fishes maybe directly disturbed, leading to stress, injury, and mortality, as a result of sampling [37]. Stress and mortality resulting from handling as well as confinement has been well documented in fishes [38–41]. However, it appears that no comparison has been made in terms of stress or mortality experienced by fishes sampled via seining or fyke netting so a data-based decision for optimal sampling method in terms of stress and mortality is not possible at this time. Nonetheless, in the field, mortality of fishes sampled via seining was noticeably higher when seining than when using fyke nets. Also, mortality during seining seemed to be higher at oxbows with greater amounts of aquatic vegetation as fish became entrapped in the vegetation and were handled for a longer period of time than in oxbows with less vegetation. Additionally, for both methods, mortality was noticeably higher as water temperatures increased later in the sampling period, as has been found by many other studies [42]. Therefore, it may be preferable to use fyke netting in oxbows with large amounts of aquatic vegetation, and to sample early in the morning (when water temperatures are lowest) to reduce potential stress and mortality.

## Conclusion

In summary, the monitoring of oxbows, both before and after restoration, is necessary to understand impacts of restoration on the fish community present within the oxbow. We must understand how restoration impacts the fish community to improve management practices that maximize benefits for target species. Based on results from deploying 3 mini-fyke nets for 24 hours in wetlands ranging from 130 m$^2$ to 2200 m$^2$ in surface area, and results obtained by seining the entire wetland area, we determined that fyke netting and seining performed equally well in sampling oxbow fish communities, and better than minnow trapping or backpack electrofishing. Therefore, we recommend seining and fyke netting as best sampling practices for oxbow fish communities. However, when choosing between these gear types, consideration must be given to amount of time needed to conduct sampling, oxbow habitat characteristics, oxbow morphology, potential habitat disturbance generated by sampling, and potential stress and mortality created by sampling with either method.

## Supporting information

**S1 Table. Abundance values for fish species captured in this study, including number of individuals in total and by sampling method, and its overall relative abundance (percent of total individuals).** Values are based on 11 sampling events for each gear type in each wetland (18 May–11 June).
(XLSX)

## Acknowledgments

We thank our field technicians, S. Grinstead and C. Wood, for the many hours spent in the field collecting the data presented within this manuscript. We also thank the various collaborating entities and individuals who have made this project possible through support of the research efforts. Collaborators for this project include C. McKinney, K. Wilke, A. Kenney, J. Olson, D. Weissenfluh, Iowa Soybean Association, Syngenta Crop Protection LLC, The Nature Conservancy, and the U.S. Fish and Wildlife Service. Finally, we thank the many landowners whose support was vital for the success of this project.

## Author Contributions

**Conceptualization:** Dylan M. Osterhaus, Samuel S. Leberg, Timothy W. Stewart.

**Data curation:** Dylan M. Osterhaus.

**Formal analysis:** Dylan M. Osterhaus, Audrey McCombs.

**Funding acquisition:** Clay L. Pierce.

**Investigation:** Dylan M. Osterhaus, Samuel S. Leberg.

**Methodology:** Dylan M. Osterhaus, Samuel S. Leberg, Timothy W. Stewart, Audrey McCombs.

**Project administration:** Dylan M. Osterhaus.

**Resources:** Clay L. Pierce.

**Writing – original draft:** Dylan M. Osterhaus.

**Writing – review & editing:** Dylan M. Osterhaus, Samuel S. Leberg, Clay L. Pierce, Timothy W. Stewart, Audrey McCombs.

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
