## [Decision Letter · Decision Letter 0]

16 Aug 2022

PONE-D-22-18618Comparison of sampling methods for small oxbow wetland fish communitiesPLOS ONE

Dear Dr. Osterhaus,

Thank you for submitting your manuscript to PLOS ONE.The manuscript has improved greatly, but we feel that additional revision is needed. Please see comments below.Therefore, we invite you to submit a revised version of the manuscript that addresses the points raised during the review process.

We look forward to receiving your revised manuscript.

Kind regards,

Stefano Larsen

Academic Editor

PLOS ONE

Journal Requirements:

"We thank our field technicians, S. Grinstead and C. Wood, for the many hours spent in the field collecting the data presented within this manuscript. We also thank the various collaborating entities and individuals who have made this project possible, either through funding or support of the research efforts. Collaborators for this project include C. McKinney, K. Wilke, A. Kenney, J. Olson, D. Weissenfluh, Iowa Soybean Association, Syngenta Crop Protection LLC, The Nature Conservancy, and the U.S. Fish and Wildlife Service. Finally, we thank the many landowners whose support was vital for the success of this project".

Additional Editor Comments:

Dear Dr Osterhaus,

I have now received comments from the two reviewers that assessed a previous version of the manuscript.

Both reviewers agreed that the revised version is greatly improved, and I agree with this view.

Before I am able to accept the manuscript for publication, you need to address a key issue raised by rev.1. This is regarding the definition of CPUE. For coherence, the reviewer proposes an update of the definition used in your study, to account of the different gears used.

Please consider the suggestions and modify the text accordingly, to the extent that you consider agreeable.

I therefore recommend minor revision at the moment.

Looking forward to receive the latest version.

Reviewers' comments:

Reviewer's Responses to Questions

**Comments to the Author**

1. Is the manuscript technically sound, and do the data support the conclusions?

Reviewer #1: Yes

Reviewer #2: Yes

2. Has the statistical analysis been performed appropriately and rigorously? 

Reviewer #1: Yes

Reviewer #2: Yes

3. Have the authors made all data underlying the findings in their manuscript fully available?

Reviewer #1: Yes

Reviewer #2: Yes

4. Is the manuscript presented in an intelligible fashion and written in standard English?

Reviewer #1: Yes

Reviewer #2: Yes

5. Review Comments to the Author

Reviewer #1: * GENERAL COMMENTS

The study reported in the manuscript PONE-D-22-18618 ("Comparison of

sampling methods for small oxbow wetland fish communities") compared

four sampling methodologies, two active and two passive gears, with

regard of effectiveness in sampling fish communities of small wetland

oxbows.

I have reviewed a former draft of the MS. The Authors have reworked

the paper, and they have considered and applied the suggestions of the

former reviewers. As a result, the MS has improved a lot. It is a well

written paper, contains relevant results and suggestions for

monitoring fish communities in small oxbows. I would like to highlight

the merit of the Discussion, which has valuable parts regarding the

practical use of the each gear in the field.

However, one issue have remained in the text that I do not agree

with. See my SPECIFIC COMMENTS below for the details, where I have

made a suggestion (an easy conceptual correction) to solve that issue.

Please consider my suggestion or another change in order to make your

paper more clear and exact.

In my opinion, the MS is publishable provided that the authors correct

the issue in question. I have some small notes as well that need to be

revised; please refer to the SPECIFIC COMMENTS.

* SPECIFIC COMMENTS

L140-141 Please correct the notes for the unit of measure of

conductivity. The correct form has a capital "S", which stands for

"Siemens".

L151-152 The sentence starting "Future studies..." maybe would be in a

better place in the Discussion. But, I see reason why you have written

that here. So, it can also remain here, but it makes a small

"stylistic" dissonance because this sentence evaluates.

L166-172 This is the issue that I do not agree with. As far as I am

concerned, Catch Per Unit Effort (CPUE) is an applicable way to

standardize data sampled only with the *same* sampling

methodology. The sampling effort is basically gear-specific (e.g.,

electrofishing time, exposure time, the number of fyke nets used, and

so on), so it can not be comparable between gears. Consequently, your

CPUE ("the number of fish collected per 100m2 of oxbow surface area")

as it is now, is not a real CPUE. On the other hand, your data need to

be standardized to control for the size differences of the

oxbows. Hence, I understand the reason why you scaled your raw catch

data to unit surface area; it is correct. But to be precise in the use

of concepts, I do recommend changing the response variable CPUE to

"catch per unit surface area" (CPUS) or "catch per unit area" (CPUA),

and leaving the CPUE. In summary, if I were you, I would highlight the

need for scaling the raw data to control for the size differences of

the oxbows, instead of reasoning with "the ability to sample the

entire fish community within an oxbow". In fact, dividing by the area

does not provide information on the entire fish community, so I am

afraid I cannot accept your argument. Please reconsider this part of

your paper. I think that the small correction I suggested would make

the text more accurate and precise.

L189 Maybe the word "levels" is not necessary because readers that are

less familiar with linear models or the jargon of statistics may not

understand what "levels" mean. You sampled each oxbow only once, so

your factor 'oxbow' in the models have 11 levels each with one

repetition. Therefore, averaging over the levels eventually means

averaging over the oxbows, if I am not mistaken. An other option can

be giving a little bit more explanation on what "levels" means.

L 320-321 Please consider changing the word "sites" to "oxbows".

Reviewer #2: I reviewed an earlier version of this manuscript and made the following over-arching recommendations, as well as some more minor recommendations. Given the revisions, I recommend acceptance.

“The manuscript needs to remove and greatly revise discussion of stress to fish, habitat disturbance, and ease of use. These objectives were neither qualitatively or quantitatively measured (e.g., time for deployment in terms of person/hours, percentage of disturbance to substrate and vegetation per unit area) nor were available literature used to support the assertions (i.e., fish stress following electrofishing versus net-based methods is well documented). Much of the discussion on stress, habitat disturbance, and ease of use is subjective and speculative. Either address by providing data, use the literature, or restrict the manuscript to quantitative comparisons of the fish assemblage and metrics. These comparisons are valuable and are publishable. Below, I have recommendations for revision. Some of these are substantial.”

The authors discussion of ease of implementation, interactions with habitat, and apparent mortality/stress are greatly improved. Number of visits and hours spent per visit are very important considerations, as is disturbance during sensitive life history stages.

The authors also did address minor recommendations. For example, the nonlinearity, non normality of the fish data were addressed, test statistics were added, and figure 1 was improved.

6. PLOS authors have the option to publish the peer review history of their article (what does this mean?). If published, this will include your full peer review and any attached files.

Reviewer #1: No

Reviewer #2: No

---

## [Author Response · Author response to Decision Letter 0]

25 Oct 2022

Journal Requirements:

All requirements have been checked and corrections have been made where necessary.

"We thank our field technicians, S. Grinstead and C. Wood, for the many hours spent in the field collecting the data presented within this manuscript. We also thank the various collaborating entities and individuals who have made this project possible, either through funding or support of the research efforts. Collaborators for this project include C. McKinney, K. Wilke, A. Kenney, J. Olson, D. Weissenfluh, Iowa Soybean Association, Syngenta Crop Protection LLC, The Nature Conservancy, and the U.S. Fish and Wildlife Service. Finally, we thank the many landowners whose support was vital for the success of this project".

Amended funding statement has been included in the cover letter and all financial mentions have been removed from the manuscript.

Corresponding author is now Timothy Stewart (twstewar@iastate.edu)

Requested statement and IACUC permit information has been added within the methods section.

6. Please review your reference list to ensure that it is complete and correct. If you have cited papers that have been retracted, please include the rationale for doing so in the manuscript text, or remove these references and replace them with relevant current references. Any changes to the reference list should be mentioned in the rebuttal letter that accompanies your revised manuscript. If you need to cite a retracted article, indicate the article’s retracted status in the References list and also include a citation and full reference for the retraction notice. Additional Editor Comments:

References have been reviewed and an error was found with reference #30 being placed prior to #27, this has been fixed. Additionally, a reference was added to an article recently published pertaining to oxbow restoration monitoring and Topeka Shiner recovery which is highly relevant to this study (Osterhaus et al. 2022).

Dear Dr Osterhaus,

I have now received comments from the two reviewers that assessed a previous version of the manuscript.

Both reviewers agreed that the revised version is greatly improved, and I agree with this view.

Before I am able to accept the manuscript for publication, you need to address a key issue raised by rev.1. This is regarding the definition of CPUE. For coherence, the reviewer proposes an update of the definition used in your study, to account of the different gears used.

Please consider the suggestions and modify the text accordingly, to the extent that you consider agreeable.

I therefore recommend minor revision at the moment.

Looking forward to receive the latest version.

All suggestions made by reviewers have been addressed at this time. We thank the editor for their comments and guidance during this process.

Comments to the Author

Reviewer #1: * GENERAL COMMENTS

The study reported in the manuscript PONE-D-22-18618 ("Comparison of

sampling methods for small oxbow wetland fish communities") compared

four sampling methodologies, two active and two passive gears, with

regard of effectiveness in sampling fish communities of small wetland

oxbows.

I have reviewed a former draft of the MS. The Authors have reworked

the paper, and they have considered and applied the suggestions of the

former reviewers. As a result, the MS has improved a lot. It is a well

written paper, contains relevant results and suggestions for

monitoring fish communities in small oxbows. I would like to highlight

the merit of the Discussion, which has valuable parts regarding the

practical use of the each gear in the field.

However, one issue have remained in the text that I do not agree

with. See my SPECIFIC COMMENTS below for the details, where I have

made a suggestion (an easy conceptual correction) to solve that issue.

Please consider my suggestion or another change in order to make your

paper more clear and exact.

The authors greatly appreciated the constructive comments and recommendations provided in the first round of review and agree that the manuscript has been greatly improved as a result of these comments and recommendations.

In my opinion, the MS is publishable provided that the authors correct

the issue in question. I have some small notes as well that need to be

revised; please refer to the SPECIFIC COMMENTS.

* SPECIFIC COMMENTS

L140-141 Please correct the notes for the unit of measure of

conductivity. The correct form has a capital "S", which stands for

"Siemens".

The S has been capitalized in each usage.

L151-152 The sentence starting "Future studies..." maybe would be in a

better place in the Discussion. But, I see reason why you have written

that here. So, it can also remain here, but it makes a small

"stylistic" dissonance because this sentence evaluates.

We agree that this sentence creates minor dissonance within the paragraph, however, we believe that the best fit for this sentence is in the location that it is currently placed. We attempted to find a better location within the discussion but could not find a placement that improved flow.

L166-172 This is the issue that I do not agree with. As far as I am

concerned, Catch Per Unit Effort (CPUE) is an applicable way to

standardize data sampled only with the *same* sampling

methodology. The sampling effort is basically gear-specific (e.g.,

electrofishing time, exposure time, the number of fyke nets used, and

so on), so it can not be comparable between gears. Consequently, your

CPUE ("the number of fish collected per 100m2 of oxbow surface area")

as it is now, is not a real CPUE. On the other hand, your data need to

be standardized to control for the size differences of the

oxbows. Hence, I understand the reason why you scaled your raw catch

data to unit surface area; it is correct. But to be precise in the use

of concepts, I do recommend changing the response variable CPUE to

"catch per unit surface area" (CPUS) or "catch per unit area" (CPUA),

and leaving the CPUE. In summary, if I were you, I would highlight the

need for scaling the raw data to control for the size differences of

the oxbows, instead of reasoning with "the ability to sample the

entire fish community within an oxbow". In fact, dividing by the area

does not provide information on the entire fish community, so I am

afraid I cannot accept your argument. Please reconsider this part of

your paper. I think that the small correction I suggested would make

the text more accurate and precise.

This is an excellent point and very well laid out. We agree with the reviewer that CPUE is not appropriate and is really not what were calculating. Therefore, we have made edits to the manuscript and figures where needed to change from CPUE to CPUA which is a much more relevant definition and accurate to what we were calculating.

L189 Maybe the word "levels" is not necessary because readers that are

less familiar with linear models or the jargon of statistics may not

understand what "levels" mean. You sampled each oxbow only once, so

your factor 'oxbow' in the models have 11 levels each with one

repetition. Therefore, averaging over the levels eventually means

averaging over the oxbows, if I am not mistaken. An other option can

be giving a little bit more explanation on what "levels" means.

We agree with the reviewer that the usage of “levels” here is unnecessary and have made the appropriate change.

L 320-321 Please consider changing the word "sites" to "oxbows".

Site and Sites have been changed to Oxbow and Oxbows

Reviewer #2: I reviewed an earlier version of this manuscript and made the following over-arching recommendations, as well as some more minor recommendations. Given the revisions, I recommend acceptance.

“The manuscript needs to remove and greatly revise discussion of stress to fish, habitat disturbance, and ease of use. These objectives were neither qualitatively or quantitatively measured (e.g., time for deployment in terms of person/hours, percentage of disturbance to substrate and vegetation per unit area) nor were available literature used to support the assertions (i.e., fish stress following electrofishing versus net-based methods is well documented). Much of the discussion on stress, habitat disturbance, and ease of use is subjective and speculative. Either address by providing data, use the literature, or restrict the manuscript to quantitative comparisons of the fish assemblage and metrics. These comparisons are valuable and are publishable. Below, I have recommendations for revision. Some of these are substantial.”

The authors discussion of ease of implementation, interactions with habitat, and apparent mortality/stress are greatly improved. Number of visits and hours spent per visit are very important considerations, as is disturbance during sensitive life history stages.

The authors also did address minor recommendations. For example, the nonlinearity, non normality of the fish data were addressed, test statistics were added, and figure 1 was improved.

We thank the reviewer for their constructive feedback and appreciate their time and effort.

---

## [Editor Report · Decision Letter 1]

2 Nov 2022

Comparison of sampling methods for small oxbow wetland fish communities

PONE-D-22-18618R1

Dear Dr. Stewart,

We’re pleased to inform you that your manuscript has been judged scientifically suitable for publication and will be formally accepted for publication once it meets all outstanding technical requirements.

Kind regards,

Stefano Larsen

Academic Editor

PLOS ONE

Additional Editor Comments (optional):

Dear Authors,

The revised version of the manuscript has clearly integrated all comments and corrections proposed by the reviewers. In particular, the issue regarding the definition of CPUE/CPUA has been addressed. The figures were updated, and minor edits also included.

I therefore can recommend the manuscript for publication.

Best wishes
---

## [Editor Report · Acceptance letter]

9 Nov 2022

PONE-D-22-18618R1 

Comparison of sampling methods for small oxbow wetland fish communities 

Dear Dr. Stewart:

I'm pleased to inform you that your manuscript has been deemed suitable for publication in PLOS ONE. Congratulations! Your manuscript is now with our production department. 

Kind regards, 

on behalf of

Dr. Stefano Larsen 

Academic Editor

PLOS ONE